# Affect and Cognitive Closure in Students—A Step to Personalised Education of Clinical Assessment in Psychology with the Use of Simulated and Virtual Patients

**DOI:** 10.3390/healthcare10061076

**Published:** 2022-06-09

**Authors:** Maciej Walkiewicz, Bartosz Zalewski, Mateusz Guziak

**Affiliations:** 1Department of Psychology, Medical University of Gdańsk, 80-210 Gdansk, Poland; 2Department of Psychology of Individual Differences, Diagnosis and Psychometric Research, SWPS University of Social Sciences and Humanities, 03-815 Warsaw, Poland; bartosz.zalewski@swps.edu.pl; 3Faculty of Medicine, Medical University of Gdańsk, 80-210 Gdansk, Poland; mateusz.guziak@gumed.edu.pl

**Keywords:** affect, interview, clinical reasoning, clinical competence, diagnosis, simulation training, patient simulation, psychology, clinical psychology, students, education, teaching

## Abstract

Introduction: Since there was no general model of competencies to determine a successful clinical assessment, we based our study on the many skills that are needed to perform one. We analysed students’ learning performance based on inner determinants, such as affect and cognitive closure, with the use of two teaching methods (i.e., simulated patient (SP) or virtual patient (VP)). Methods: The sample comprised 56 fifth-year clinical psychology students. The need for closure (NFC) and efficacy in fulfilling the need for closure (EFNC) were measured using standardised questionnaires. The authors’ VP and SP tools were used to teach and measure the effectiveness of learning psychological interview techniques and clinical reasoning. Clinical interview skills included building contact with the patient, gathering important information and making mistakes. Clinical reasoning skills were divided into eight dimensions for the assessment of mental health. Results: Affect and cognitive closure are important psychological variables in anticipating and developing interview and clinical reasoning skills for psychology students. The simulated patient was more effective for interview skills, while the virtual patient was a beneficial teaching tool for most clinical reasoning skills. Virtual patient training was a useful teaching method for students with a low EFNC, probably because it provided a stable and strong structure. Simulated patient training was effective for people with a high EFNC, presumably because it allowed them to build on their advanced structuring skills. Conclusions: Affect and cognitive closure can be used to identify students’ learning abilities to provide a more personalised education. The results of the present study may be useful for evaluating different teaching methods, monitoring their effectiveness and enhancing students’ performance.

## 1. Introduction

Questions concerning the inner determinants of the acquisition of diagnostic skills are surprisingly lacking in publications on the psychological profession [1]. Traditional training in clinical diagnosis stresses the need to build a cooperative relationship with the patient, collect data [2,3] and analyse them to integrate complex information and social interactions involving social exposition and emotional burden [4]. 

To conduct a successful diagnostic interview in psychology, students must learn to regulate their emotions and recognise those of others. Therefore, *affect*—as part of more advanced states, such as emotions, feelings, preferences, moods and traits [5,6]—plays an important role in learning [7,8,9]. A small, temporary change in affective state can significantly impact the performance of a challenging cognitive task requiring flexible thought or a clear solution [10]. Affect is usually classified as either positive (PA) and negative (NA) [11]. The former facilitates higher problem-solving and clinical reasoning abilities [12], while the latter modulates cognitive functions. Positive affect is known to decrease cognitive inhibition in response to emotionally marked material [13]. In learning clinical skills, higher PA can improve verbal fluency [14,15] and reduce interference between competing responses, which results in faster reaction time between congruent and incongruent stimuli [16].

The cognitive–motivational aspects of reasoning are expressed through the need for cognitive closure (NFC) and efficacy in fulfilling the need for closure (EFNC) [17]. The NFC is defined as the desire for an answer to a given problem [18]. In the context of clinical reasoning in psychology, a high NFC is associated with: a more limited search for information before a decision is made [19]; higher ratings of confidence after a decision is made [20]; a stronger preference for familiar choices instead of new options; and a tendency to stereotype [21,22,23]. Efficacy in fulfilling the need for closure refers to the individual’s ability to reach swift decisions and create structure in life. The level of EFNC is determined by the extent to which individuals can use different styles of information processing according to their NFC [24].

Since there was no general model of competencies to determine a successful clinical assessment, we based our study on the skills that are known to be required: clinical interview and reasoning skills.

*Clinical interview skills* involve building contact with the patient, gathering important information and making mistakes [25]. *Clinical reasoning skills* refer to: (1) The ability to recognise patients’ negative and positive aspects of functioning [26]; (2) The reactance of a patient. A reactant patient is easily provoked and responds oppositional to perceived external demands [27]; (3) The coping style of a patient, which is an enduring personality trait when a person confronts new problematic situations. We distinguish two styles: externalising (impulsive, stimulation-seeking, extraverted) and internalising (self-critical, inhibited, introverted) [28]; (4) The stage of change of a patient, which represents a person’s readiness for psychological change, defined as a period of time and set of tasks needed for movement to the next stage [28]; (5) The cognitive errors in clinical reasoning: (a) Confirmation bias: the tendency to look for confirming evidence to support a diagnosis rather than look for disconfirming evidence to refute it [29]. (b) Overconfidence bias: a universal tendency to believe we know more than we do [29]. (c) Multiple alternative bias: a multiplicity of options on a differential diagnosis may lead to significant conflict and uncertainty [29]. (d) Overpathologisation bias: not explicitly mentioned in medical literature, however, very similar to other biases distinguished, e.g.,: premature closure, representativeness restraint, search satisficing [30], ascertainment bias, diagnosis momentum [29] and focusing effect [31]; (6) Adequacy of collected data refers to the quality and completeness of collected diagnostic data presented by the SP and VP; (7) The general quality of assessment, which refers to the general measurement of quality of diagnosis.

We applied two teaching methods to investigate a wide range of diagnostic competencies. One involved direct contact, while the other was computer mediated; this enabled the measurement of dependent variables at two levels of interaction. The first method—simulated patient (SP)—demands significant involvement, since the learner has to perform a complex diagnostic activity, bridge cognitive biases, formulate diagnostic interventions and more [32,33]. The second—virtual patient (VP)—entails a lower cognitive–emotional burden because the student does not have direct contact with the patient. At the same time, they can be provided with information on the steps they have to take [34].

The present study complements the literature by assessing the value of SP and VP in the acquisition of both interview and clinical reasoning skills. The goal of the study was to analyse students’ learning performance based on affect and cognitive closure with the use of the simulated patient (SP) and virtual patient (VP).

## 2. Materials and Methods

**Material**. The sample comprised 56 fifth-year clinical psychology students (86% female) at the University of Social Sciences and Humanities, Warsaw, Poland (average age *M* = 28.48; *Me* = 24; *SD* = 8.53). Participation in the study was one of the options in a mandatory student internship. Respondents could withdraw from participation at any time. In total, 69 students applied for the project: SP N = 38; VP N = 31. N = 56 took part in the study after the two-day training course. SP initial session N = 25; SP final session N = 24. VP initial session N = 31; VP final session N = 27. The number of participants who completed the entire project from the beginning to the end was N = 51.

**Procedure**. The study was carried out between September 2017 and May 2018. The participants took part in a two-day training course (12 h) on the process of clinical diagnosis. The independent variables NFC and EFNC were measured. Four SP sessions and four VP sessions were conducted. Half of the randomly selected participants took part in SP sessions, and the other half took part in VP sessions. Each session consisted of completing the SP or VP procedure and writing a clinical diagnosis. The goal of the initial sessions was to assess the primary level of students’ *diagnostic skills* (dependent variables). For the SP sessions, the level of clinical interview and reasoning skills were assessed by competent judges, as were the participants’ diagnoses of the patients. For the VP sessions, the level of clinical interview skills was assessed by the VP computer program, while the participants’ clinical reasoning skills were assessed by competent judges. The second and third sessions were used for training. The SP participants received feedback from a competent judge who observed the interview, and the VP students received feedback generated by the VP program. The final level of the participants’ diagnostic skills (dependent variables) was assessed during the final sessions (see Figure 1).

**The independent variables** were measured using two questionnaires. Affect was measured using the Positive and Negative Affect Schedule-Trait (PANAS) [11,35,36]. This questionnaire contains 30 items on two subscales that assess a person’s positive affective states (i.e., active, alert, attentive, determined, enthusiastic, excited, inspired, interested, proud and strong) and negative ones (i.e., afraid, ashamed, distressed, guilty, hostile, irritated, nervous, scared and upset) [37,38,39,40]. The PANAS uses a Likert-type scale, with answers ranging from *very slightly* or *not at all* (1) to *extremely* or *very much* (5).

To measure NFC, we used Webster and Kruglanski’s (1994) NFC scale. Respondents rated the 32 items on a 6-point scale (from 1 = *completely disagree* to 6 = *completely agree*). This scale has good psychometric properties [41].

Efficacy at fulfilling the need for closure was measured using nine items (high and low levels) and a 6-point scale (from 1 = *completely disagree* to 6 = *completely agree*). A higher score implied a higher level of EFNC. This scale has good psychometric properties [24].

**The dependent variables**:

*Clinical interview skills.* These were measured using the VP authors’ computer tool, which consists of short recordings of a person playing the role of the patient in a diagnostic psychological consultation lasting 20–25 min. The recordings were based on case studies of real patients with severe mental disorders using the Keyes and Lopez dimension model, which is based on a description of positive and negative aspects of the patient’s functioning, resulting in four types of profiles, from healthy persons to patients with more severe mental disorders [42,43]. Each prototype was described for both female and male roles. In Appendix A, we provide examples of the individual patients played. Each prototype included a description of the problems reported by the patient, the duration and severity of symptoms, a short history of the patient’s personal and professional life and the environment from which s/he came from, including a history of romantic relationships and relations with loved ones. Finally, a description of the behaviour during the interview and the objectives of the relevant therapy were added. The type of disorder was described by indicating the dominant symptoms, the nosological diagnosis of the disorder, the level of reactance and readiness for change and the style of coping. In the ‘healthy’ profile, there was no individual disorder or extensive symptoms; the simulant talked about her/his greatest concerns, being significantly disturbed by the behaviour of a loved one. The participants’ task was to recognize that this particular SP does not manifest disorders but talks about another person’s mental difficulties.

The participants watched nine successive fragments of a patient interview, and after each one, they chose one of two diagnostic interventions. The program enables 512 combinations of selection paths arranged in a decision tree. In the second and third VP training sessions, the participants received feedback after each decision (Appendix B).

For the SP, interview skills were measured live by competent judges on scales based on: (1) building contact with the patient; (2) gathering important information; and (3) making mistakes. The SPs session lasted 50 min, which is the standard time for a psychological interview. The simulated patients were experienced academic teachers and practising clinicians (i.e., not actors). They developed their roles during 10 h workshops and played the same ones with all participants (Appendix A).

*Clinical reasoning skills* were measured by a tool (designed by the authors) that estimates the way one formulates diagnostic hypotheses after the interview with an SP or a VP (Appendix A). The competent judges evaluated whether students applied the following dimensions of patient characteristics in their clinical reasoning: (a) *negative and positive aspects of a patient’s functioning*; (b) *reactance*; (c) *coping style*; (d) *stage of change*; (e) *cognitive errors*; (f) *adequacy of collected data*; and (g) the *quality of assessment* (Appendix D, Appendix E, Appendix F). The competent judges were experienced, academic teachers. They and the participants were anonymous.

**Statistical analysis**. IBM SPSS 25 software was used for data analysis. Pearson correlations were conducted to investigate the relationships in the data.

**Ethics statement**. The present study was conducted in accordance with the guidelines of the Ethical Review Board at the University of Social Sciences and Humanities in Warsaw, Poland, and was reviewed and approved (decision 23/IV/11-12). Written consent was obtained from all participants.

## 3. Results


*Interview skills*


*Building contact with the patient* presents: a low negative correlation with the *PA* in the final VP session, a low negative correlation with *EFNC* in the initial SP and the final VP session.

*Mistakes*presents a low negative correlation with *EFNC* in the initial SP session.


*Clinical reasoning skills*


*Negative aspects of patient’s functioning* presents a low positive correlation with *EFNC* in the initial VP session.

*Positive aspects of patient’s functioning* presents a low negative correlation with *NFC* in the final VP session.

*Reactance* presents a negligible positive correlation with *PA* in the final SP session.

*Coping style* presents a low positive correlation with *PA* in the final VP session.

*Confirmation bias* presents a low positive correlation with *PA* in the initial VP session, a low positive correlation with *EFNC* in the initial VP session.

*Overconfidence bias* presents a negligible positive correlation with *EFNC* in the initial SP session.

*Multiple alternative bias* presents a low negative correlation with *PA* in the initial SP session, a negligible positive correlation with *NFC* in the final SP session.

*Overpathologisation bias* presents a negligible positive correlation with *PA* in the initial VP session, a low positive correlation with *NFC* in the initial SP, a low positive correlation with *NFC* in the final VP, a low positive correlation with *EFNC* in the final SP, a low positive correlation with *EFNC* in the initial VP session, a moderate positive correlation with *EFNC* in the final VP session.

*Adequacy of collected data* presents a low positive correlation with *PA* in the initial VP, a negligible positive correlation with *EFNC* in the initial VP session (see Table 1).

## 4. Discussion

We analysed students’ learning performance based on inner determinants, such as affect and *cognitive closure*, with the use of two teaching methods (i.e., simulated patient (SP) or virtual patient (VP)). The results showed that SP was more effective for all interview skill variables and VP for most clinical reasoning skill variables.

In the SP task, participants with high PA and EFNC were experiencing a significant amount of stress. A high level of PA creates a natural tendency to regulate tension through positive emotions and thus satisfaction at having completed a task. The need to *build contact* is neglected as a result. The highly structured VP program did not allow the participants to generate their own responses. Therefore, it may be detrimental for people with a high EFNC because they build structure naturally on their own [24].

The SP training provided expressive feedback and immediate responses to participants’ errors, so that high-EFNC participants made fewer *mistakes*. Participants with a low EFNC could not structure the interview with the SP and, therefore, made more and more *mistakes* over time. Those *mistakes* were probably based on a momentary but illusory sense of confidence and resulted in the participants offering incorrect advice and misinterpreting the patient’s behaviour.

The *need for cognitive closure* was not associated with any interviewing skill for either SP or VP because it involves cognitive processes (clinical reasoning) rather than information gathering (interviewing). Another possible explanation is that the interview framework (contact/structure/mistakes) fulfils the NFC, and the participants conducted the interview regardless of their NFC level, implying that it does not inhibit interview learning.

*Clinical reasoning skills* appeared to be much more complex variables and generated more results. Participants had to analyse patient responses in several dimensions to formulate key diagnostic hypotheses.

During SP, higher NA levels were associated with less effective diagnoses of the patient’s *reactance*, perhaps because the participants were more self-centred, mood regulating and lacking the flexibility they needed to recognise that the patient may have had internal reasons for resistance or that the diagnostician was building an unfavourable therapeutic alliance, which is consistent with the *reactance* definition [27].

During VP training, participants with higher NFC may have narrowed their processing of psychopathology and formed their hypotheses adequately. They may also have realised they should choose the most significant data and quickly abandoned processing other information that may have led to clear, unambiguous conclusions.

Identifying *coping style* is a relatively simple diagnostic skill because different ones have clear indicators. Participants with higher PA were better able to recognise them.

*Stage of change* may not have been noticed or considered as a significant source of information because it does not appear in any dimension to describe the patient’s characteristics. We supposed that because our respondents were at a very early stage of their education; they had not paid any attention to it. Perhaps they believed that patients seeking psychological counselling would be motivated to change. Experienced doctors know that patients’ motives can be more complex; indeed, people experiencing emotional distress are often ambivalent about seeking help at all.

The majority of *cognitive errors* were eliminated with our training. This is a promising result because it suggests that both SP and VP tools are effective in reducing mistakes and strengthening critical reasoning skills.

High-EFNC participants stopped committing *confirmation bias*, which suggests that they reduced their cognitive distortions through the training. The VP participants recognised more and more alternative interpretations of the decision paths and made more and more composite diagnoses throughout their training. This effect did not occur in SP because it allowed participants to collect complex and multi-layered data immediately.

High-EFNC participants learned how to assess more complex structures and the multidimensional nature of the patient in SP reducing *overconfidence bias*. They also reduced their *multiple alternative biases*. This meant that they built more complex and multifaceted hypotheses after training. On the other hand, those with a low NFC built too many alternatives and too many hypotheses and were unable to verify them with any reliability.

*Overpathologisation bias* refers to an excessive focus on the negative aspects of a patient’s functioning [31]. The participants who focused overly on learning how to diagnose psychopathology during training likely gave too much attention to this aspect. Because it is a highly structured tool, the VP program seemed not to help students with a high NFC, while a less structured approach (i.e., SP) did. In theory, we would expect the opposite for a complex task such as *clinical reasoning* [24]. Participants with a high EFNC in the SP group began to pathologise patients excessively. They quickly found a clear structure for organising diagnostic information and did not allow other alternative explanations. A high level of EFNC did not support participants in building complex, multifaceted diagnoses. The opposite was the case for participants with a low EFNC who perhaps need more time to structure information and generate hypotheses. Their low synthesising skills allow them to build hypotheses from a smaller but they are more diverse amount of data, covering both positive and negative aspects of functioning.

Participants with higher PA in the VP group were more able to assess the information at the beginning of training but stopped thereafter, perhaps because they became more critical. Moreover, the previously described mechanism seems to be confirmed by the results obtained from the emotional burden scales.

The VP tool provided only a limited amount of information about the patient, but the participants in the SP group could ask questions freely and obtain more data as a result. They were then able to propose hypotheses based on a wider range of information than was the case with the VP.

## 5. Conclusions

Affect and cognitive closure are important psychological variables in anticipating and developing interview and clinical reasoning skills among psychology students. SP training seems to be more effective for all interview skills, whereas VP training can be a beneficial teaching tool for most clinal reasoning skills. VP training is a useful teaching method for students with a low EFNC, probably because it provides a stable and strong structure. SP is effective for people with a high EFNC, presumably because it allows them to build on their high structuring skills. Both affect and cognitive closure might help in identifying students’ learning abilities and matching their competencies to provide a more personalised education.

### Limitations

The study was conducted at one university and involved fifth-year clinical psychology students, limiting its applicability to the general population of a trainee. All results were based on interviews and clinical reasoning conducted with simulated and virtual patients. Changes in emotional distress appear to be amenable to investigation not only by self-report but also by physiological measures, but this was not the case in the present study.

## Figures and Tables

**Figure 1 healthcare-10-01076-f001:**
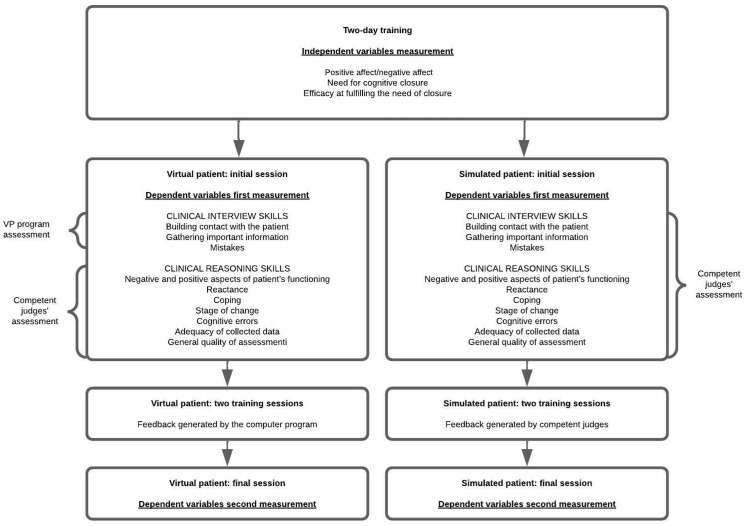
The study procedure.

**Table 1 healthcare-10-01076-t001:** Relationships between affect, need and efficacy at fulfilling the need for closure of diagnosticians and psychological assessment skills learnt through the use of simulated and virtual patients.

	Positive Affect—PA (PANAS)	Negative Affect—NA (PANAS)	Need for Cognitive Closure (NFC)	Efficacy at Fulfilling the Need for Closure (EFNC)
**Interview Skills**				
1. Building contact with the patient	VP/F r(32) = −0.421; *p* = 0.016			SP/I r(56) = −0.337; *p* = 0.011 VP/F r(32) = −0.485; *p* = 0.005
2. Gathering important information				
3. Mistakes				SP/I r(47) = −0.315; *p* = 0.031
**Clinical Reasoning Skills**				
1.1. Negative aspects of patient’s functioning				VP/I r(41) = 0.314; *p* = 0.046
1.2. Positive aspects of patient’s functioning			VP/F r(43) = −0.390; *p* = 0.010	
2. Reactance		SP/F r(47) = 0.291; *p* = 0.047		
3. Coping style	VP/F r(42) = 0.315; *p* = 0.042			
4. Stage of change				
5.1. Confirmation bias	VP/I r(48) = 0.310; *p* = 0.032			VP/I r(41) = 0.441; *p* = 0.004
5.2. Overconfidence bias				SP/I r(55) = 0.290; *p* = 0.031
5.3. Multiple alternative bias	SP/I r(51) = −0.303; *p* = 0.031		SP/F r(48) = 0.289; *p* = 0.046	
5.4. Overpathologisation bias	VP/I r(48) = 0.298; *p* = 0.040		SP/I r(46) = 0.308; *p* = 0.037 VP/F r(43) = 0.343; *p* = 0.024	SP/F r(49) = 0.305; *p* = 0.033 VP/I r(28) = 0.395; *p* = 0.038 VP/F r(21) = 0.556; *p* = 0.009
6. Adequacy of collected data	VP/I r(47) = 0.306; *p* = 0.036			VP/I r(51) = 0.298; *p* = 0.034
7. Quality of assessment				

SP—simulated patient; VP—virtual patient; I—initial session; F—final session.

## Data Availability

The data presented in this study are available on request from the corresponding author.

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
