# Peer review of "Affect and Cognitive Closure in Students—A Step to Personalised Education of Clinical Assessment in Psychology with the Use of Simulated and Virtual Patients"

_healthcare, 2022, doi:10.3390/healthcare10061076_

Round 1

Reviewer 1 Report

  1. Explain the  Keyes and Lopez dimension model
  2. Explain some of your statistics such as the relationship between effect, need and efficacy
  3. Explain 'coping style' as a diagnostic skill
  4. 212  Explain the adequacy of cognitive errors
  5. 254 The conclusions  should be written in paragraphs, rather than numbered point form..

Author Response

We would like to thank the Reviewer for the comments and positive review of our manuscript. We hope the revision made our text more suitable for publication.

Point 1: Explain the  Keyes and Lopez dimension model.

Response 1. We added information about the Keyes and Lopez model in the ‘dependent variables’ section.

Point 2: Explain some of your statistics such as the relationship between effect, need and efficacy.

Response 2: We considered this suggestion by adding a paragraph explaining the results in the "Results" section.

Point 3: Explain 'coping style' as a diagnostic skill

Response 3: In the 'Introduction', we added descriptions of ‘coping style’ and other ‘clinical reasoning skills’. We hope the concept is much clear now.

Point 4: 212  Explain the adequacy of cognitive errors

Response 4: We added a description of  ‘Clinical reasoning skills’ such as ‘Cognitive errors’ in the Introduction.

Point 5: 254 The conclusions should be written in paragraphs, rather than in numbered point form.

Response 5:  We rewrote the conclusions into narrative form.

Reviewer 2 Report

  • Statistical characteristics in the section 2.1. Is ti about the cohort of participants enrolled to the study, if yes then it is quite bizarre to represent descriptive data with SD to N parameter.
  • SPSS – please mention version
  • Results chapter require major editing. To present results only in one Table is very bizarre and incomprehensible for a common reader. In addition, the results must be presented in the relevant chapter rather than in Discussion. Discussion should be focused on your data comparison and comparative analysis to other studies with a few statements regarding significance of your study. Please amend accordingly.
  • The diverse bias endpoints measured and mentioned in your Table must be presented in Methods chapter. Please describe what each bias mean, why is it relevant to be included into the study (i.e. rationale)
  • Discussion can’t be reviewed at this stage due to it requires major editing as mentioned in my previous statement.
  • Conclusions should be presented in a narrative form. This is not a data from meta-analysis paper.

Author Response

We kindly thank the Reviewer for the insightful review and valuable comments and feedback. The text was reorganized according to the Reviewer’s suggestions. We hope that the changes made the manuscript more suitable for publication. Herby, we respond directly to the Reviewer's comments:

Point 1: Statistical characteristics in the section 2.1. Is ti about the cohort of participants enrolled to the study, if yes then it is quite bizarre to represent descriptive data with SD to N parameter.

Response 1: We changed statistical characteristics in the section 2.1.

Point 2: SPSS – please mention version

Response 2: We mentioned the SPSS version.

Point 3: Results chapter requires major editing. To present results only in one Table is very bizarre and incomprehensible for a common reader. In addition, the results must be presented in the relevant chapter rather than in the Discussion. Discussion should be focused on your data comparison and comparative analysis to other studies with a few statements regarding the significance of your study. Please amend accordingly.

Response 3: We would like to thank you very much for this comment. We edited the ‘Results’paragraph.

Point 4: The diverse bias endpoints measured and mentioned in your Table must be presented in the Methods chapter. Please describe what each bias means, and why is it relevant to be included into the study (i.e. rationale)

Response 4: We corrected this oversight. We tried to simplify various descriptions to make the text concise and missed this topic. In the 'Introduction' section, we developed an excerpt "bias" and other variables about 'Clinical reasoning skills'. Adding this description in the 'Introduction' section instead of 'Methods' seemed more suited to the context of the text. Nevertheless, we hope that the changes made the paragraph more readable and acceptable to the Reviewer.

Point 5: Discussion can’t be reviewed at this stage due to it requires major editing as mentioned in my previous statement.

Response 5: We hope that the added content in or the previous point will make the ‘Discussion’ section easier for the reader to understand the context.

Point 6: Conclusions should be presented in a narrative form. This is not a data from meta-analysis paper.

Response 6: The conclusions were changed into narrative form.

Reviewer 3 Report

The article title and abstract are appropriate.
The purpose of the article and its significance is stated clearly.
The study methods are sound and appropriate.
The writing is clear and concise.
The conclusions or summary are accurate and supported by the content.
The article is of interest to members of the education research community.

Author Response

We would like to thank the Reviewer for the positive revision of our manuscript. The whole text has been edited by proofreading services for the academic authors - Proof-Reading-Service.com. We hope that our manuscript is suitable for publication.

Reviewer 4 Report

The topic is interesting. However, the paper is poorly written and needs to be improved. I recommend the author(s) make the suggested changes indicated below and re-submit the article for consideration. Some detailed comments are given below.

  1. In the material section, it is said “M = 28.48”, does this refer to age?
  2. In the research results section, only one table without any text description or explanation?
  3. It is unreasonable that the page number of appendix has more than the content of this article.
  4. I don't think this article is currently considered for acceptance for publication in the way it is presented.

Author Response

We kindly thank the Reviewer for the insightful review. The text has been reorganized according to the recommendations. We hope that the changes made the manuscript more suitable for publication. Herby, we respond detailed to the Reviewer's comments:

Point 1: In the material section, it is said “M = 28.48”, does this refer to age?
Response 1: Yes, we corrected the information on the patient's age in the ‘Methods’ section.

Point 2: In the research results section, only one table without any text description or explanation?
Response 2: We would like to thank the Reviewer for this comment. We corrected this oversight and added the description of the results. 

Point 3: It is unreasonable that the page number of the appendix has more than the content of this article.
Response 3: We realize that it might be unreasonable that the page number of the appendix has more than the content of the article. Six appendixes refer to the methodological part of our study. Due to the complicated procedure, we decided not to include the detailed description in the manuscript itself. A detailed description would be excessively long. At the same time, to maintain the highest possible scientific reliability, we decided to attach "documentation" of the conducted research. We also believe that due to the originality of the research tools it was valuable to attach them as complete tools we provide might help to create further VPs and SPs for academic teachers and researchers in medical education.

Point 4: I don't think this article is currently considered for acceptance for publication in the way it is presented.
Response 4: We would like to thank you for the valuable feedback and hope that our revision made the text more suitable for publication.

Round 2

Reviewer 2 Report

Authors have responded to my previous comment. The Discussion section requires major editing. Please do not repeat the results of your study in discussion. This chapter requires thorough comparative analysis of your data to the data presented on other studies and presenting ideas of why the data is different or similar to other studies. Please amend.

In addition, the title of the paper is conducted with 2 sentences. This is unusual. It will be better of authors can restructure a title in one sentence.

Author Response

We would like to express our gratitude to the Reviewer for the constructive revision of our manuscript. In response to feedback, we reconstructed a title and revised the Discussion section. We've also made some minor English language changes. We are hopeful that our manuscript is now suitable for publication.

Reviewer 4 Report

Thank you for making the appropriate corrections to my comments.

The manuscript has been much improved and is in a nice condition now.

I considered that the modifications made improve the quality of the manuscript.

Author Response

We would like to express our gratitude to the Reviewer for the constructive revision of our manuscript. The Discussion section has been updated further. We've also made some minor English language changes. We are hopeful that our manuscript is now suitable for publication.